# Advances in the Application of Electrostatics in Agriculture: A Review from Macroscale Spray Engineering to Microscale Plant Biostimulation

**DOI:** 10.3390/mi16111285

**Published:** 2025-11-14

**Authors:** Jie Cao, Zhelin Jin, Juan He, Guizhang Ju, Letian Mi, Yang Gao, Rui Lei, Guanggui Cheng

**Affiliations:** 1School of Mechanical Engineering, Jiangsu University, Zhenjiang 212013, China; 2112003010@stmail.ujs.edu.cn (J.C.); 2112303018@stmail.ujs.edu.cn (Z.J.); 2222403065@stmail.ujs.edu.cn (G.J.); 2222403074@stmail.ujs.edu.cn (L.M.); 2222503155@stmail.ujs.edu.cn (Y.G.); leirui@ujs.edu.cn (R.L.); 2Changzhou City Urban Flood Control Engineering Administration, Changzhou City Water Conservancy Bureau, Changzhou 213000, China; hejuan6776@163.com

**Keywords:** electrostatic technologies, spatial-scale, electrostatic spraying, plant biostimulation, precision agriculture

## Abstract

Electrostatic technology has emerged as a crucial tool for sustainable agricultural development due to its multifunctional characteristics. However, systematic and specialized investigations into its mechanism of action and application principles across diverse agricultural scenarios remain insufficient. Here, this review innovatively constructs a spatial scale classification framework and categorizes it into macroscale spray engineering and microscale plant biostimulation. At the macroscale, electrostatic spraying leverages charged droplets’ properties (high surface charge density, strong electrostatic interaction, enhanced adsorption) to improve canopy deposition efficiency and reduce agrochemical drift losses. At the microscale, electrostatic fields induce electron/ion directional movement, providing non-contact stimulation to regulate plant physiological processes such as seed germination and nutrient uptake. We systematically summarize the latest research progress in electrostatic spraying and electrostatic biostimulation, and further compare them in terms of their fundamental mechanisms, targets, and stages of technological development. Finally, the current limitations and challenges for each technology are overviewed and the forward perspective for the efficient application of electrostatics in agriculture are outlined. This review provides theoretical references and technical guidelines for the application research of electrostatic spraying and electrostatic biostimulation, holding significant importance for promoting the standardized development of electrostatic technology in sustainable and precision agriculture.

## 1. Introduction

The global agricultural system is under mounting pressure from the intertwined challenges of population growth, environmental degradation, and resource scarcity [1,2]. In addition, agriculture is highly susceptible to the impacts of climate change, with rising temperature extremes, droughts, and pest outbreaks posing threats to crop yields and stability [3,4,5]. In response, the paradigm of sustainable agriculture has gained global urgency. Central to this transformation is the development of high-efficiency, low-input technologies capable of optimizing resource use while minimizing environmental footprint [6,7,8,9]. Among these, precision agriculture has emerged as a pivotal approach, which enables spatially and temporally optimized interventions by tailoring inputs such as water, fertilizers, and pesticides to the actual needs of crops [10,11,12,13,14,15]. This not only improves input-use efficiency and crop performance but also reduces waste, pollution, and production costs [16,17,18,19].

Current precision agriculture technologies remain limited by mechanical constraints, high costs, and insufficient adaptability to complex field conditions. Particularly, challenges persist in achieving targeted delivery of agrochemicals across diverse canopy architectures, as well as in stimulating crop performance at the physiological or cellular level without external chemical inputs [20,21,22]. These gaps underscore the need for novel physical tools and mechanisms that can function across scales—from field-level deployment to direct plant-level modulation—with minimal invasiveness and maximal specificity [23,24,25,26]. Electrostatics, as a fundamental physical phenomenon characterized by controllable force fields and charge interactions, offers a promising yet under-explored solution space. Its ability to manipulate droplets, particles, and biological structures through electric forces opens avenues for precise agricultural interventions [27,28].

### 1.1. Fundamental Principles of Electrostatics and Its Agricultural Potential

Electrostatics is the branch of physics that describes how charged entities interact via Coulomb forces—the fundamental interactions that govern the attraction or repulsion between particles. At the heart of electrostatic phenomena is the electric field (E), which is a vector field surrounding a charge and exerts a force on other charges within its influence. Unlike mechanical contact forces, electrostatic fields act at a distance, enabling non-contact manipulation of particles or materials. The behavior of materials in these fields is largely determined by their electrical properties: conductors (such as metallic electrodes) allow free movement of charges and are used to generate or transfer electric fields [29,30,31], while insulators (such as polymer surfaces or plant tissues) restrict charge mobility and are often polarized rather than directly charged [32,33,34]. Two essential physical mechanisms enable the use of electrostatics in real applications. The first is corona discharge, which involves the ionization of air molecules near a high-voltage electrode, producing ions that can be transferred to droplets, particles, or surfaces, thereby charging them [35,36]. The second is electrostatic induction, which allows neutral but polarizable objects, like plant leaves or pollen grains, to develop induced charge distributions when exposed to an electric field. This results in attractive mirror-image forces, even without direct contact or conduction [37].

Fundamental physical principles can be harnessed to develop solutions for complex agricultural systems, where the effects are largely governed by the intrinsic characteristics of the interacting targets and agents. Notably, these agricultural targets and agents often exhibit three key properties: small size, mobility, and responsiveness to electric fields. Droplets, pollen grains, fungal spores, dust particles and even plant tissues can all interact with electrostatic fields in predictable ways [38,39,40]. Coulomb forces can be used to steer charged droplets or particles toward specific plant organs, overcoming gravity, wind, or structural barriers. Electrostatic fields themselves can act as energy stimuli, modulating cell membrane potentials or triggering biochemical responses. Electrostatic adhesion enables particles to bind efficiently to irregular plant surfaces, increasing deposition efficiency and reducing off-target loss [41,42,43]. These capabilities suggest that electrostatics is not merely compatible with agricultural operations, but also suited to augment their efficiency and specificity. Electrostatic technologies bring multiple inherent benefits to agriculture. They reduce agrochemical drift and enhance deposition on target crop surfaces in pesticide spraying via directional charge control; act as a non-thermal and non-chemical stimulus to avoid thermal damage and chemical residues, thus aligning with green and organic agriculture; and feature low energy consumption and scalable designs, making them suitable for both large-scale farming and on-site localized precision interventions [44,45,46,47,48,49]. These advantages support various applications, including electrostatic-assisted pollination, electrostatic spraying for accurate pesticide and fertilizer delivery, electrostatic seed selection and grading, electrostatic harvesting and separation, electrostatic promotion of crop growth, and electrostatic pest and disease control [50,51,52,53,54,55]. Crucially, electrostatics provides unique cross-scale utility [56]. In macroscopic terms, it enables centimeter to millimeter scale manipulation of droplets in canopy spraying systems for crop protection interventions. Microscopically, it facilitates subcellular modulation of plant physiology, affecting ion channels, enzymatic activity, and stress signaling pathways [57]. Starting from the fundamental physical effects underlying electrostatic applications in agriculture, this review emphasizes the potential of electrostatics to address multi-scale challenges by bridging macroscale spray engineering and microscale electrostatic biostimulation. The inherent scalability of electrostatic effects from mechanical transport to molecular stimulation underpins the spatial-scale-based classification framework of this review.

### 1.2. A Multi-Scale Framework for Reviewing Electrostatic Applications in Agriculture

Over the past decades, considerable progress has been made in exploring electrostatic technologies for agricultural applications. Existing reviews have thoroughly addressed electrostatic spraying systems, particularly their mechanisms, nozzle designs, and roles in improving agrochemical deposition [58,59]. Separately, a growing number of studies have begun to examine electrostatic stimulation of plant physiological responses, exploring its potential in enhancing germination and growth. These studies have significantly advanced their respective subfields. However, few attempts have been made to gain a coherent understanding of how electrostatics based on common physical principles works in different agricultural scenarios.

To overcome the fragmentation in existing literature, this review introduces a unified classification framework based on the spatial scale of electrostatic interaction. We distinguish two major domains: (1) macroscale applications, where electrostatics serves as an engineering tool to manipulate charged droplets and particles in the canopy space, typified by technologies such as electrostatic spraying and air-assisted delivery systems; (2) microscale applications, where electrostatic fields act as non-contact stimuli to modulate cellular and biochemical processes in plants, encompassing seed activation, growth promotion, and bioelectric signaling. This spatial-scale perspective bridges physical engineering and biological regulation, offering a conceptual lens to integrate electrostatic agriculture-applications under a cohesive theoretical structure.

Following the spatial-scale framework, the remainder of the review is structured accordingly. Section 2 focuses on the macroscopic dimension, reviewing the physical mechanisms of droplet charging, advances in electrostatic spraying equipment, and their system-level implementations. Section 3 transitions to the microscale dimension, exploring the mechanistic basis and agricultural applications of electrostatic biostimulation, including emerging self-powered systems. Finally, Section 4 synthesizes insights from both domains, comparing their fundamental mechanisms, targets, stages of technological development, current limitations and challenges, and proposing an outlook for cross-scale integration. This review aims to summarize the transition of electrostatics from well-established engineering practices to emerging biological frontiers, while revealing the potential coherence of electrostatics as a multi-scale platform for sustainable agricultural innovation.

## 2. Macroscale Electrostatic Spraying from Charging Mechanisms to System Integration

### 2.1. Physical Basis and Charging Mechanisms of Electrostatic Spraying

The physical principles of electrostatic spraying are grounded in the interaction between electrically charged fluids and external electric fields [59,60,61]. The principle involves intricate electrohydrodynamic behaviors that govern how charged droplets are generated, transported and deposited. Understanding the dynamics of jet formation, droplet charging and field-induced breakup is essential for optimizing the design of electrostatic spraying engineering.

Electrostatic spraying relies on the use of strong electric fields to induce the atomization and charging of liquid jets, enabling the generation of charged droplets with enhanced deposition effect [62]. One of the earliest systematic analyses of this phenomenon was conducted by R. L. Hines in 1966 [63]. Through carefully designed experiments using a hemispherical electrode and a planar collector (Figure 1a), Hines demonstrated that a fluid issuing from a small orifice under a high-voltage electric field forms a stable cone-jet configuration followed by droplet breakup. By combining photographic analysis of the jet morphology (Figure 1b) with electric current and mass flow measurements, the relationships linking the droplet charge-to-mass ratio (q/m) to fluid properties (viscosity, conductivity, surface tension, density) and the applied field strength was established. As shown in Figure 1c, the q/m could reach values on the order of 3.3 × 10^−6^ C/g under typical operating conditions, sufficient to overcome gravitational and inertial forces for efficient electrostatic deposition. Furthermore, Anestos et al. investigated the underlying sources of charge in electrostatic sprays [64]. They believe that the primary mechanism of droplet charging does not originate from external corona discharge, but rather from the disruption of a charged liquid surface during jet breakup. Through a series of comparative experiments using the modified collection apparatus of paint sprays and electrometers, they demonstrated that the efficiency of droplet charging is strongly dependent on the electrical conductivity of the fluid. There exists an optimal conductivity range (approximately 2 μmho/cm) within which the specific charge of droplets is maximized.

Faraji et al. conducted a detailed experimental study on the role of electrical conductivity in determining spray modes under high-voltage fields [59]. As shown in Figure 1d, six distinct modes including dripping, multispindle, spindle, microdripping, oscillating jet and unstable cone jet were observed with high-speed imaging. The results (Figure 1e) demonstrate that higher electrical conductivity lowers the voltage required to initiate cone-jet formation. This occurs because higher conductivity enables faster charge relaxation and greater accumulation of surface charges, which intensifies the normal electric stress acting on the liquid meniscus. Moreover, mode selection exhibits a nonlinear dependence on the conductivity–voltage parameter space, offering a robust framework for fluid formulation and process optimization in electrospray applications. On the other hand, Chen and Deng developed a multiphysics modeling framework to investigate droplet breakup dynamics under coupled electrostatic and aerodynamic forces [60]. Figure 1f,g illustrates how electric stress elongates the mother droplet, while aerodynamic shear promotes surface instability and detachment of daughter droplets. Quantitative results (Figure 1h) reveal that both electric current and charge-mass ratio β increase with rising applied voltage, confirming the enhancement of charge transport and atomization efficiency. These findings provide mechanistic insights for optimizing droplet generation in high-performance electrostatic atomizers, significantly advancing the development and technological upgrading of electrostatic spray systems.

Over decades of technological advancement and mechanistic investigation, the charging mechanisms underlying electrostatic spraying have been gradually elucidated. Yu et al. provided a systematic summary of the three principal charging modes employed in electrostatic spray systems: contact charging, corona charging, and induction charging [65]. In contact charging, a high-voltage electrode is placed in direct electrical connection with the working fluid. As the fluid is atomized through the nozzle, droplets acquire charge via conduction from the liquid body itself. Corona charging utilizes a sharp high-voltage electrode to generate a localized ionized region through air breakdown. The resulting ions adhere to the surfaces of atomizing droplets, effectively imparting charge. Lastly, induction charging leverages the electrostatic induction effect by placing electrodes near, but not in contact with, the liquid stream. An external field induces spatial charge separation, leading to the accumulation of opposite charges on the liquid jet and adjacent electrode structures. The investigation of electrostatic spraying mechanisms is foundational to the spraying technology, which significantly advanced the development of spraying systems and spraying engineering.

### 2.2. Design and Optimization of Electrostatic Atomization Nozzles

The mechanisms underlying electrostatic spraying are now relatively mature, while the performance and applicability of electrostatic spraying systems in agriculture rely heavily on the structural configuration of atomization nozzles [66,67,68]. Recent years have witnessed significant advancements in nozzle design, targeting improvements in spray precision, droplet size, charge efficiency, and adaptability to various crop environments [69,70,71].

To enhance deposition effect and reduce pesticide waste, modern nozzle designs focus on optimizing electrode configuration, airflow dynamics and charging mechanisms. Lin et al. developed a waist-shaped induction electrode-based electrostatic nozzle to improve vertical distribution uniformity in protected horticultural crops [72]. The system employs a waist-shaped electrode design and by keeping the plate-connecting portion, enlarges the equivalent liquid film area to ultimately generate a higher induced charge (Figure 2a). Through field deployment in tomato greenhouses, they demonstrated that the optimized nozzle design achieved a more focused spray distribution pattern, reduced lateral drift and minimized pesticide waste on non-target areas. Comparative field results confirmed that the electrostatic spray coverage at the canopy’s central vertical plane significantly increased compared to conventional nozzles, especially under variable ambient airflow conditions. Khatawkar et al. designed a backpack-mounted air-assisted electrostatic sprayer for small-scale farms and orchard use [73]. The system employs an integrated nozzle-electrode assembly powered by a high-voltage DC supply (Figure 2b) to efficiently charge droplets via electrostatic induction. This core is supplemented by an electric ducted fan and liquid feed control, allowing for on-demand spraying with adjustable flow rates and voltage. Field trials revealed notable improvements in deposition effect across vertical crop surfaces, particularly on the undersides of leaves. In addition, the nozzle could maintain stable operation under different wind speeds due to enhanced electrostatic attraction forces.

Focusing on droplet size control and atomization efficiency, Gao et al. proposed a novel low-frequency ultrasonic electrostatic nozzle incorporating a double-resonator structure (Figure 2c) [67]. The system combines acoustic and electrostatic atomization mechanisms by using a primary and secondary resonance cavity to fragment the liquid column into fine droplets. An electrostatic induction ring positioned around the Laval tube exit further imparts surface charge. Experimental evaluations showed that the system achieved droplet sizes as small as 7.8 μm, with most conditions yielding droplets below 25 μm, particularly under high gas pressure and electrostatic voltage. The design proved suitable for aeroponic cultivation systems, where fine droplet size and high adhesion are critical. Patel et al. proposed an enhanced air-assisted nozzle designed to optimize air-liquid mixing and electrostatic charging for orchard applications [61]. The system employs a coaxial flow design with high-voltage and ground electrodes (Figure 2d) to generate a strong axial electric field. The interaction between axial airflow and radial liquid feed enables the atomization of charged droplets with consistent size and directionality. Comparative bench tests demonstrated that this nozzle achieved a uniform spray cloud, significantly improving the deposition effect in complex canopy structures. Gao et al. further explored the design of a high-voltage electrostatic ultrasonic atomization nozzle with a focus on root-level deposition in aeroponic systems [42]. Their nozzle integrates a Laval tube for supersonic airflow, an ultrasonic excitation module for fine droplet formation and a contact charging mechanism at the nozzle inlet (Figure 2e). Evaluation results of the nozzle’s performance indicated that using the copper electrode at 12 kV, 0.4 MPa pressure and a 1.75 m spray distance resulted in maximum droplet adhesion on the root system. Numerical simulations using COMSOL further confirmed that electrostatic forces improve droplet coverage and penetration into dense root networks. These findings validate the feasibility of electrostatic nozzles in precision nutrient delivery for aeroponics.

Collectively, innovations in electrode configuration, acousto-electric hybridization and airflow integration have significantly optimized the performance and applicability of electrostatic nozzles. In response to the multifaceted demands of sustainable precision agriculture, modularity, systematization, and integration have become emergent trends in electrostatic spraying.

### 2.3. Integration of Intelligent Spraying Systems with Electrostatic Modules

The integration of electrostatic modules with intelligent spraying systems aiming to improve droplet deposition effect, canopy penetration and automation performance. Recent studies have explored novel structural designs and control strategies to synergistically combine electrostatic charging with precise, intelligent application platforms [74,75,76,77].

To address the challenge of uneven droplet deposition in boom sprayers, Liu et al. proposed an inductive electrostatic spraying system based on an embedded closed-electrode structure [75], as shown in Figure 3a,b. The system employs embedded copper plates within the nozzle to form a sealed electrostatic induction module, thereby avoiding direct exposure of electrodes and minimizing safety risks. Field trials demonstrated that increasing the charging voltage from 0 to 14 kV led to significant improvements in droplet deposition across different canopy positions (Figure 3c), with the greatest enhancement observed on the lower back side of leaves. This indicates that the system has the potential to promote wrap-around deposition via electrostatic attraction, which in turn leads to improved pesticide coverage in shaded or obstructed regions. Amaya et al. focused on the design of an insulated induction electrode for air-assisted orchard sprayers, aiming to overcome droplet loss caused by turbulent airflow [76]. As shown in Figure 3d–f, the electrostatic module consists of an insulated central electrode ringed by high-voltage elements, which generates a stable induction field around the nozzle. The field trials revealed that under electrostatic conditions, both adaxial and abaxial droplet coverage significantly improved, especially in the middle and lower canopy layers. The results highlight the system’s ability to guide charged droplets toward fruit tree canopies and reduce drift loss through enhanced electrostatic attraction. Zhao et al. developed a UAV-based contact-charging electrostatic spraying system [77]. As shown in Figure 3g–i, the system employs a direct-contact charging mechanism, where a high-voltage generator transfers negative charges to the liquid inside the tank, enabling in-flight droplet charging without external electrodes. The UAV integrates power, control and spraying modules in a compact configuration (Figure 3h), achieving coordinated control of trajectory and spray output. Field evaluations demonstrated that electrostatic spraying increased the effective spray width by 1 m and raised the average droplet deposition density to 19.7 droplets/cm^2^. Visual comparison in Figure 3j showed that electrostatic spraying produced a denser, more uniform spray cloud than conventional methods. Additionally, as indicated by the sampling results in Figure 3k, the charged droplets exhibited a more concentrated and symmetrical deposition profile, validating its application potential in precision agriculture.

### 2.4. Emerging Applications of Electrostatic Spraying in Environmental and Material Control

Building upon the core technologies of nozzles and intelligent systems, the utility of electrostatic spraying is now expanding into novel, non-conventional agricultural domains, notably in environmental remediation and functional material fabrication [78,79,80]. Oh et al. (2025) developed an integrated electron beam–electrostatic spray system for high-efficiency removal of fine particulates from industrial flue gas streams [79]. As shown in Figure 4a, the system combines gas pretreatment, electron irradiation and electrostatic spraying of reactive solutions. Performance evaluation (Figure 4b) showed that Na_2_SO_3_ spray significantly enhanced dust and NO_X_ removal compared to water, especially at higher voltages. Zhang et al. investigated the synergistic effect of ammonia scrubbing and electrostatic spraying for flue gas purification from straw combustion [81]. As depicted in Figure 4c, a modular electrostatic unit was installed to charge the atomized spray directly at the nozzle outlet. Dust removal efficiencies across different straw types (rice, wheat, corn) are presented in Figure 4d. Without charge (0 kV), removal efficiencies were consistently below 20%. However, ammonia-treated sprays achieved dust removal rates exceeding 50% at 8 kV, especially for wheat straw, which evidenced the substantial benefit of electrostatic enhancement in gas purification applications.

Yang et al. expanded the application of electrostatic spraying to material science, focusing on the development of biodegradable chitosan (CS) films using atmosphere-controlled high-voltage electrospray (AHES) [82]. As schematically shown in Figure 4e, this method applied up to 100 kV across a nozzle–collector system, with nitrogen or oxygen serving as propellant gases to control the plasma environment during film formation. The resulting CS films exhibited notable electrical conductivity and electrostatic adhesion (Figure 4f). Specifically, 100 kV AHES-treated films achieved conductivity increases of over 100% and resistivity reductions of 50% compared to untreated films, supporting their potential application in smart packaging. The antimicrobial performance of these films was validated using inhibition zone tests and colony counting methods. The inhibition zones for *S. aureus* and *E. coli* (Figure 4g) expanded from 22 mm in untreated films to 30 mm and 27 mm, respectively, in 100 kV-treated samples, indicating the enhanced antibacterial efficacy due to increased NH_3_^+^ group density on CS chains. Additionally, Figure 4h presents muskmelon slices stored using CS-based film packaging. After three days, AHES-treated CS films combined with polyethylene (PE) maintained significantly better preservation quality compared to untreated or control groups, demonstrating its superior moisture barrier and antimicrobial performance.

In summary, macroscale electrostatic spraying has evolved into a mature and validated technology, enabling precise and efficient agrochemical delivery through optimized nozzle design, charge control, and intelligent system integration. Beyond conventional spraying for crop protection, its utility has expanded into environmental engineering and material functionalization. However, these applications primarily operate through physical transport mechanisms. In contrast, emerging evidence suggests that electrostatic fields may also modulate biological processes within plants. The next part explores this microscale dimension, where electrostatics acts not as a delivery tool but as a biophysical stimulus capable of influencing plant physiology at the cellular and molecular levels.

## 3. Microscale Electrostatic Biostimulation and Self-Powered Agricultural Systems

When the spatial scale shifts from the canopy to the cellular level, the static electric field transitions from being a conveyance tool to a potent biophysical stimulant. This part turns its focus to the micro-scale realm, where electric fields directly interact with plant tissues and cellular processes [83,84]. We will first delve into the fundamental mechanisms underpinning plant responses to electric fields, from altered membrane permeability to enhanced enzymatic activity. Subsequently, we will review empirical evidence of its efficacy in promoting seed germination and regulating plant growth. Finally, we will explore the cutting-edge paradigm of self-powered systems based on TENGs, which promise to unlock the full potential of this bio-electrical interaction for sustainable and precision agriculture.

### 3.1. Mechanistic Insights into Electrostatic Stimulation of Plant Physiology

The remarkable capacity of electrostatic fields to enhance plant growth and stress resilience has garnered significant interest in recent years. A substantial body of experimental evidence derived from diverse crops demonstrates positive outcomes in seed germination, photosynthetic efficiency and biomass accumulation [85,86,87,88,89]. However, the underlying biophysical and biochemical mechanisms responsible for these benefits remain elusive and lack a consolidated theoretical framework.

Zhang et al. investigated the impact of pulsed electric field (PEF) treatment on the physicochemical behavior of chlorophyll aggregates [85]. As shown in Figure 5a, chlorophyll solubility followed a nonlinear trend in response to varying electric field intensity and temperature, with optimal dispersion achieved at intermediate field strengths (around 16.8 kV/cm). Radial distribution function plots (Figure 5b) showed a decrease in the intensity and sharpness of g(r) peaks after PEF exposure, indicating reduced molecular packing density. Molecular dynamics simulations illustrated a time-dependent loosening of aggregate structure over 100 ns (Figure 5c), confirming the disaggregation of chlorophyll molecules into smaller oligomers. These results suggest that electrostatic fields alter pigment conformation and enhance bioavailability through molecular-level destabilization and solvation. Zhang et al. obtained further mechanistic insights by examining the germination response of brown rice under moderate electric field exposure [90]. They proposed that electric stimulation induces reversible electroporation of cell membranes, thereby enhancing permeability to water and ions. As illustrated in Figure 5d, this electroporative effect leads to controlled production of reactive oxygen species, which serve as secondary messengers to activate antioxidant defense pathways. Specifically, moderate electric field-treated seeds showed elevated activity of superoxide dismutase (SOD), catalase (CAT), and peroxidase (POD), along with increased accumulation of antioxidant metabolites such as flavonoids and total phenolics. These changes collectively form a coordinated antioxidant system, thereby alleviating oxidative damage induced by reactive oxygen species and consequently promoting faster and more robust germination.

Li et al. introduced a self-powered agricultural system based on ambient energy harvesting [84], wherein mechanical stimuli such as wind and rainfall are converted into electrostatic potential via triboelectric nanogenerators (TENGs). As shown in Figure 5e, the system enabled uniform application of electric fields (up to 600 V/cm) to both aerial and root zones of pea seedlings grown in hydroponic conditions. The study revealed that exposure to the electrostatic field markedly accelerated shoot growth during the early vegetative stage (Figure 5f), which was accompanied by a significant increase in the uptake of essential mineral nutrients, including potassium, calcium, manganese and iron (Figure 5g). These changes were attributed to electric field-induced modulation of ion transport mechanisms and membrane polarization effects. Moreover, the concentration of nitrogen oxides rises from 0 to 3.8 mg L^−1^ within 1 min in a sealed container (Figure 5h), and the result of the test paper shows that the nitrate concentration in the water exceeds 25 mg L^−1^, indicating the production of nitrogen fertilizer. Furthermore, microscopic and biochemical analyses indicated that plants subjected to field stimulation exhibited higher levels of nitrate assimilation, chlorophyll biosynthesis, and photosynthetic efficiency. The authors believe that the enhanced nutrient mobilization and redox regulation may stem from electric field-facilitated electron transfer processes and activation of intracellular signaling cascades that promote resource allocation toward growth.

These findings demonstrate that electrostatic fields can regulate plant physiology through multiple interrelated mechanisms, including pigment disaggregation, membrane electroporation, ion transport modulation and redox signaling activation. Although a unified conclusion has not yet been reached regarding the specific promoting mechanisms, current evidence supports the role of electric stimulation as a promising, non-chemical approach to enhance early growth, nutrient acquisition and stress resilience in plants.

### 3.2. Electrostatic Enhancement of Seed Germination and Vigor

Building on the mechanistic insights of electrostatic stimulation of plant, this section focuses on how electrostatic stimulation affects seed germination and early-stage vigor. Electrostatic fields have been shown to positively influence seed germination, vigor, and early-stage growth through non-thermal, non-invasive mechanisms that alter water uptake, membrane permeability, and metabolic activity [90,91,92,93]. In 2009, Wang et al. showed that 55 min exposure of aged rice seeds to 250–450 kV m^−1^ raised the activity index by 29%, cut electrolyte leakage by 25%, and elevated SOD, POD and CAT activities, while malondialdehyde dropped by 27%, indicating strengthened membrane integrity [91]. Afterwards, Basiry et al. applied 11.5 kV corona discharge during belt separation of barley at 6.5% moisture [37]. Sorted seeds germinated at 94.7% versus 77.3% for unsorted, a 17% gain attributed to brief, low-energy charging that accelerated water uptake and enzyme activation.

Santoyo et al. evaluated the germination and early growth responses of *Arabidopsis thaliana* and *Mammillaria mathildae* under continuous exposure to uniform electrostatic fields [92]. As illustrated in Figure 6a, the experimental setup consisted of a high-voltage power supply connected to an array of anode and cathode electrodes arranged above and below the seed substrate. Seeds were exposed to varying field intensities ranging from 0.1 to 0.8 V cm^−1^. Germination rates, measured weekly over a four-week period, showed a consistent increase under electric field conditions compared to the control, with optimal enhancement observed at 0.2 V cm^−1^ (Figure 6b). Additionally, the phenotypic analysis of *Arabidopsis thaliana* plantlets at 6 weeks revealed substantial improvements in biomass accumulation and leaf expansion under electrostatic treatment, with the 0.2–0.4 V cm^−1^ group exhibiting the most vigorous growth (Figure 6c).

Pelesz et al. investigated the effects of high static electric fields on *Avena sativa* and *Raphanus sativus* seeds [93]. As shown in Figure 6d, they used a grounded shield to isolate control and treatment groups, applying a high-voltage DC field (~185 kV m^−1^) to the test samples. After several days of cultivation, radish seeds exposed to the electric field showed visibly denser sprouting and accelerated early development (Figure 6(ei)). Quantitative assessment revealed that radish sprout height increased by approximately 8% by the final day, with over 3 times more seeds germinating initially compared to controls (Figure 6f). In contrast, oat seeds exhibited a slight inhibition in germination and stem height under the same conditions (Figure 6(eii)), with the final maximum sprout height reaching about 92% of the control value (Figure 6g). This inhibitory effect on oats may be attributed to the high field intensity (185 kV m^−1^) approaching the dielectric threshold of air, potentially inducing subtle oxidative stress or membrane-level disturbances that delay germination, especially under the nutrient-free viscose substrate used in the study.

### 3.3. Electrostatic Promotion of Plant Growth and Physiological Performance

Beyond the seedling stage, electrostatic fields can continue to modulate plant growth and metabolism, offering a route to enhance photosynthesis, nutrient uptake, and stress resilience throughout the life cycle. Electrostatic fields have been shown to modulate plant morphology, nutrient transport, and physiological activities by inducing cell polarization, enhancing membrane permeability and activating metabolic pathways [94,95,96]. These effects contribute to increased crop yields and resilience, offering promising applications in sustainable agriculture.

Recent studies offer insights into how such fields improve crop performance across diverse systems and species. Oikonomou et al. using a conductive eSoil scaffold to deliver electrical stimuli to barley seedlings [44]. As illustrated in Figure 7a–d, barley seeds were pregerminated and cultured on an electroactive growth medium composed of a soil matrix integrated with carbon-based electrodes and a custom power circuit (Figure 7a). Electrical stimulation was applied continuously for five days (from day 5 to day 10), followed by a five-day post-stimulation growth phase, over a total 15-day observation period (Figure 7c). The electrical circuit (Figure 7b) ensured consistent potential delivery. Quantitative analyses revealed that electrically stimulated plants exhibited significant increases in both length and dry biomass across whole plants, shoots, and roots compared to non-stimulated controls (Figure 7d). These enhancements were attributed to improved ionic uptake and cell elongation, suggesting that electrostatic cues could modulate hormonal and metabolic pathways critical for growth regulation. Complementing this microscale approach, Kim et al. investigated the morphological outcomes of applying vertical (V) and horizontal (H) electrostatic fields to lettuce plants using mesh electrodes (Figure 7e) [94]. Their setup involved the application of 5 kV fields over a multi-day cultivation period under controlled environmental conditions. Quantitative and visual comparisons of plant phenotypes indicated substantial enlargement of leaf area and overall plant biomass in both V and H field treatments relative to control (Figure 7f). Notably, vertical fields had a more pronounced effect on canopy expansion. These results highlight the directional dependency of electrostatic stimuli, where field orientation may affect ion flux patterns, membrane polarization, and intracellular signaling cascades, ultimately contributing to enhanced morphogenesis.

Wu et al. provided mechanistic insights into how high-voltage electrostatic fields influence inorganic nitrogen uptake in cucumber [57]. Field strengths ranging from 0 to 3 kV cm^−1^ were applied to nutrient solutions, and kinetic measurements revealed that nitrate uptake capacity increased significantly with field intensity up to 1.5 kV cm^−1^. Conversely, ammonium uptake exhibited a biphasic response: a slight increase at 1 kV cm^−1^ followed by a significant decrease between 2 and 3 kV cm^−1^. These effects were attributed to electrostatic enhancement of membrane transport efficiency and feedback regulation within nitrogen assimilation pathways. Wang et al. investigated the individual and combined effects of non-thermal plasma seed treatment and electrostatic field application on Chinese cabbage (*Brassica rapa* ssp. *Pekinensis*) [96]. Compared to the untreated controls, plants exposed to a moderate electrostatic field alone achieved 1.7-fold greater height and higher leaf and root dry weights. Furthermore, when the field was superimposed on 20 s plasma-treated seeds, vitamin C, soluble sugar, and total amino acid contents rose by up to 22%, 22%, and 135%, respectively, suggesting that properly tuned electrostatic fields can serve as a biophysical catalyst for plant metabolic performance.

### 3.4. TENG-Based Self-Powered Stimulation Systems

To translate microscale stimulation into scalable agricultural practice, recent studies have introduced TENGs into the intelligent agricultural field [97,98,99,100,101]. This innovation opened new pathways for self-sustained plant stimulation systems, especially in scenarios where external power sources are either limited or entirely unavailable. TENGs convert low-frequency environmental mechanical energy including wind, rain, or human motion into high-voltage electrical output, enabling continuous, low-cost, and eco-friendly stimulation of crops. Moreover, by recovering mechanical energy from the agricultural environment to power sensors or microsystems, TENG enables self-powered real-time monitoring of the agricultural environment, demonstrating tremendous potential and promising prospects in facilitating the development of smart agriculture [102,103,104,105,106]. Recent studies have demonstrated that TENG-driven systems can significantly enhance seed germination, plant growth, and physiological performance by providing localized electric fields without the need for wired infrastructure [84,107]. These systems hold promise for integration into precision agriculture and greenhouses, offering a novel path toward low-carbon, intelligent farming.

Gao et al. developed a biodegradable, gelatin-based TENG device to harness wind energy for stimulating plant development [107]. As illustrated in Figure 8a,b, the system operates through contact-separation between triboelectric layers under airflow excitation, converting mechanical energy into pulsed high voltage (~900 V) and current (3000 nA) without external power. When pea seeds were exposed to the electric field generated by this system, significant improvements in germination and early-stage seedling development were observed. As shown in Figure 8c,d, the germination rate increased progressively over 1, 3, 5, and 7 days, reaching over 85% compared to ~60% in the control. Visual tracking of seedling emergence confirmed enhanced vigor and uniformity. Notably, the gelatin matrix of the TENG was designed to degrade post-use, ensuring full environmental compatibility. This system represents a sustainable, low-cost solution for decentralized agricultural stimulation, particularly in rural and low-resource settings.

Li et al. developed a scalable, field-deployable TENG system powered by ambient wind and rainfall to autonomously stimulate crops in controlled-environment settings [84]. As illustrated in Figure 8e,f, the device employs a dual-electrode setup integrated within a modular seedbed system, enabling uniform electric field distribution across germination trays. Controlled trials with pea demonstrated that under an electric field strength of 3 kV cm^−1^, germination potential increased by approximately 26.3% (Figure 8g), while under 400 V cm^−1^, seedling growth rate improved by about 17.9% compared to the control. Furthermore, growth uniformity and root elongation were markedly enhanced, as shown in Figure 8h, with taller, denser, and better-rooted sprouts in the treated group. The system’s modularity allows for flexible scaling across different crop types and environments, and its energy-harvesting architecture ensures autonomous operation without batteries or wired connections. The authors attributed the observed bioeffects to improved membrane permeability and enhanced physiological activities such as enzyme activation and photosynthesis, suggesting that the electrostatic field accelerated early plant development. These studies collectively demonstrate the feasibility of TENG-based self-powered systems as a novel biophysical tool for crop stimulation. By harvesting ambient mechanical energy and converting it into localized electrostatic fields, such systems enable sustainable, battery-free agricultural activation. Their modular design, environmental compatibility, and low energy threshold make them well-suited for integration into precision agriculture frameworks. TENG-driven platforms hold significant potential to support the development of smart farming technologies by enabling decentralized, sensor-linked, and environmentally adaptive electrostatic stimulation, thereby contributing to the next generation of intelligent, low-carbon agricultural systems.

This part highlighted the expanding role of electrostatics in modulating plant physiology at the cellular and molecular levels. From promoting seed germination to enhancing growth and stress resilience, electric field-based stimulation shows great potential as a non-chemical, energy-efficient tool in sustainable and precise agriculture. The development of self-powered electrostatic systems further enhances the scalability and field applicability of these techniques. Although challenges remain in mechanism and field validation, the current evidence underscores the transformative potential of microscale electrostatics as a novel paradigm for sustainable crop management.

## 4. Discussion and Outlook

### 4.1. Cross-Scale Comparative Analysis of Macroscale and Microscale Electrostatic Applications

Electrostatics in agriculture spans a broad functional and spatial spectrum, from field-scale spraying systems to subcellular plant responses. While these macroscale and microscale applications share foundational physical principles, they diverge significantly in their mechanisms of action, target objects and technological maturity.

Firstly, in terms of the mechanism of action, macroscale applications primarily employ the Coulombic force to manipulate the trajectories of externally charged particles, such as pesticide droplets. This process relies on well-defined physical events including corona charging, electrostatic atomization, and field-directed transport, all aimed at improving the precision of material delivery. In stark contrast, microscale applications involve the interaction of electrostatic fields with biological tissues at the cellular or molecular level. Here, the field itself acts as a non-invasive energy stimulus, modulating intrinsic physiological processes such as membrane potential, ion flux, and enzyme activity through mechanisms like electropermeabilization and electrophysiological signaling. Secondly, regarding their target objects, macroscale systems are designed to interact with external physical structures, such as leaves, stems, and entire canopies, operating within the centimeter-to-meter range. Conversely, microscale applications focus on internal biological components, from seed embryos and root meristems to cellular organelles and ion channels, engaging with targets at the micrometer or even nanometer scale. This fundamental distinction directly dictates their respective requirements for precision, exposure regimes, and methodologies for evaluating efficacy. Finally, in their technological maturity and validation methods, macroscale electrostatic spraying is a well-established engineering discipline. It is supported by commercialized products and standardized testing protocols, and its performance is quantitatively validated through crop yield assessments and computational simulations. However, microscale biostimulation remains a nascent and exploratory domain. Although laboratory studies show promising phenotypic and biochemical results, validation largely depends on complex biological assays like transcriptomics and metabolite profiling. The field currently grapples with a lack of standardized protocols, difficulties in scaling effects from controlled environments to the field, and an overall lower technology readiness level, highlighting its status as a frontier science.

### 4.2. Synthesized Challenges and Limitations

The macroscale and microscale applications of electrostatics in agriculture differ fundamentally in their mechanisms of action, target objects and technological maturity. These differences inevitably shape distinct sets of challenges, both in technical focus and in scientific bottlenecks.

One of the most persistent obstacles in macroscale electrostatic applications is environmental instability. Field conditions such as high humidity accelerate charge dissipation, while fluctuating wind speeds disrupt the controlled trajectory of charged droplets, severely compromising the effectiveness observed in controlled environments. This gap between laboratory efficacy and field reliability remains a major hurdle to widespread adoption. Engineering robustness and cost-efficiency further complicate deployment. High-voltage generation units and corona electrodes must withstand harsh field environments, including moisture, dust, and mechanical shock, all while ensuring user safety. Meanwhile, the process of achieving and maintaining optimal charge-to-mass ratios, which play a vital role in deposition efficiency and drift control, requires precise control and calibration, and this is often carried out under dynamic field conditions. In addition, integration with precision agriculture platforms, such as variable-rate applicators and UAVs, introduces significant system-level complexity. Challenges include real-time control logic, power management for high-voltage modules, and the fusion of electrostatic control with sensor-based decision-making.

The primary bottleneck of the microscale electrostatic applications lies in the lack of well-defined dose–response relationships. Here, “dose” refers to the magnitude and spatial distribution of electrostatic charge or field intensity delivered to plant surfaces, while “response” encompasses the resulting physiological outcomes such as oxidizing reaction or cellular uptake. The absence of quantitative correlations between these two factors hampers the optimization and predictability of microscale electrostatic interventions. Unlike chemicals or light, the biological effects of electric fields are not governed by straightforward concentration or intensity rules. Instead, outcomes depend on a complex interplay of field strength, frequency and exposure time often with poor reproducibility. To mitigate this issue, advanced experimental designs incorporating precise control of field parameters, environmental conditions, and plant physiological states are required. In addition, integrating real-time monitoring systems and computational modeling can also help improve reproducibility across studies. Biological heterogeneity adds further uncertainty. Variations in plant genotype, developmental stage, physiological status and ambient stressors all influence how plants perceive and respond to electrostatic stimulation, making it difficult to define universally effective treatment protocols. Most critically, the molecular and cellular mechanisms remain largely elusive. This mechanistic opacity limits the rational design and optimization of electrostatic biostimulation tools. In addition, environmental factors such as humidity, temperature and air ionization, as well as technical challenges in maintaining uniform field strength at microscale, may further contribute to variability in electrostatic biostimulation outcomes.

Despite their differences, both domains suffer from a lack of standardized protocols and evaluation metrics. For macroscale applications, no consensus exists on deposition measurement or drift quantification. For microscale applications, standardized indicators for plant response and health outcomes are equally lacking, hindering reproducibility and benchmarking.

### 4.3. Conclusions and Perspectives

This review proposed a unified spatial-scale framework to analyze and contextualize the diverse applications of electrostatics in agriculture. By systematically comparing macroscale canopy engineering and microscale plant biostimulation, we identify fundamental divergences in their mechanisms of action, target objects, technological maturity and associated challenges. Together, these two application spheres form the complementary pillars of electrostatic agricultural technologies. Looking ahead, a strategic and integrated research agenda is needed to bridge current knowledge gaps and accelerate field-level adoption. Several directions are especially critical:Mechanistic elucidation at the microscale. To address the lack of mechanistic clarity in electrostatic biostimulation, future studies should leverage multi-omics technologies and biophysical modeling to elucidate the complete signal transduction pathway, from initial electric field perception to downstream cellular responses. This knowledge will support parameter optimization and biological safety assessments.Technological integration and cross-scale system design. To overcome environmental instability in macroscale systems and standardization challenges in microscale applications, future platforms need to combine adaptive engineering with biological responsiveness. Intelligent systems that integrate real-time sensing, variable-rate control, and electrostatic modulation could synchronize field-level spraying with localized biostimulation.Paradigm shifts via self-powered systems. Emerging TENG technologies offer a disruptive solution to the power and scalability bottlenecks in microscale applications. TENGs can convert ambient mechanical energy into electric fields, enabling autonomous, localized plant stimulation. Key research frontiers include improving energy conversion efficiency, stability under field conditions, and integration with plant-compatible materials.Standardization and interdisciplinary collaboration. The lack of consistent testing protocols for both droplet behavior and plant physiological response hinders comparability and reproducibility. A cross-scale evaluation framework which standardizes metrics for deposition, energy input, and bioeffectiveness is urgently needed. Equally important is fostering deep and sustained collaboration across agricultural science, engineering, physics and materials science to break through current disciplinary silos, which will enable holistic innovation and accelerate the translation of fundamental research into practical, sustainable agricultural solutions.

In conclusion, electrostatics in agriculture is evolving from a collection of niche technologies into a potential integrated platform for sustainable crop production. The field now stands at a critical inflection point: moving beyond isolated use cases toward multi-scale, multi-functional systems that can simultaneously enhance productivity, resource efficiency, and ecological compatibility. By deepening our understanding of electrostatic interactions across biological and physical scales, and fostering innovative fusion between macroscale engineering and microscale biostimulation, electrostatics is poised to become a core enabling technology in the future of smart, resilient, sustainable and precision agriculture.

## Figures and Tables

**Figure 1 micromachines-16-01285-f001:**
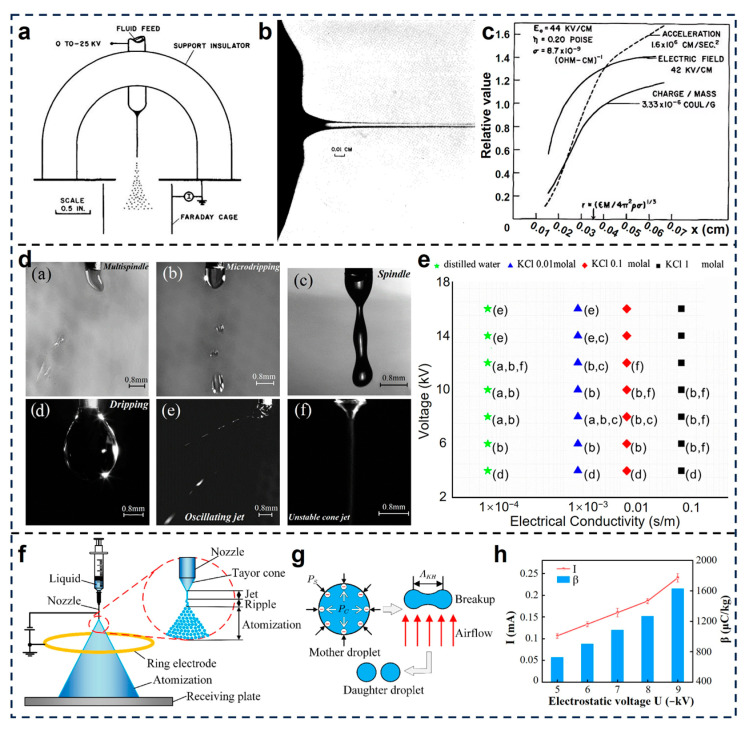
Charging mechanism of electrostatic spraying. (**a**) Experimental apparatus for measuring charging and atomization from a single jet and (**b**) the photograph of a jet [63]. (**c**) Acceleration, charge/mass ratio, and electric field as a function of the distance along the jet axis for the jet [63]. Reproduced with permission from Reference [63]. Copyright © 1966, AIP Publishing. (**d**,**e**) Electrospraying modes for different KCl solution in different applied voltage [59]; Reproduced with permission from Reference [59]. Copyright © 2017, Elsevier. (**f**) Electrostatic spray principle and (**g**) droplet breakup mechanism [60]. (**h**) Current and charge–mass ratio at different electrostatic voltages [60]. Reproduced with permission from Reference [60]. Copyright © 2022, Elsevier.

**Figure 2 micromachines-16-01285-f002:**
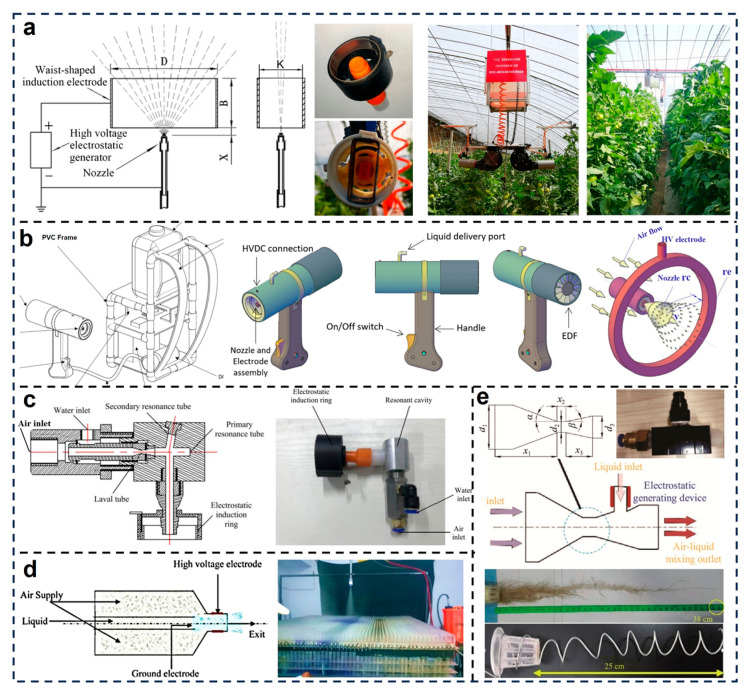
Design and optimization of electrostatic atomization nozzles. (**a**) Electrostatic spraying with waist-shaped charging devices [72]; Reproduced with permission from Reference [72]. Copyright © 2024, MDPI. (**b**) Knapsack air assisted electrostatic sprayer [73]; Reproduced with permission from Reference [73]. Copyright © 2024, Spring Nature. (**c**) Low frequency ultrasonic electrostatic atomizing nozzle with double resonators [67]; Reproduced with permission from Reference [67]. Copyright © 2022, International Journal of Agricultural and Biological Engineering. (**d**) Air-induced air-assisted electrostatic nozzle [61]. Reproduced with permission from Reference [61]. Copyright © 2017, Elsevier. (**e**) High-voltage electrostatic ultrasonic atomization nozzle [42]. Reproduced with permission from Reference [42]. Copyright © 2023, International Journal of Agricultural and Biological Engineering.

**Figure 3 micromachines-16-01285-f003:**
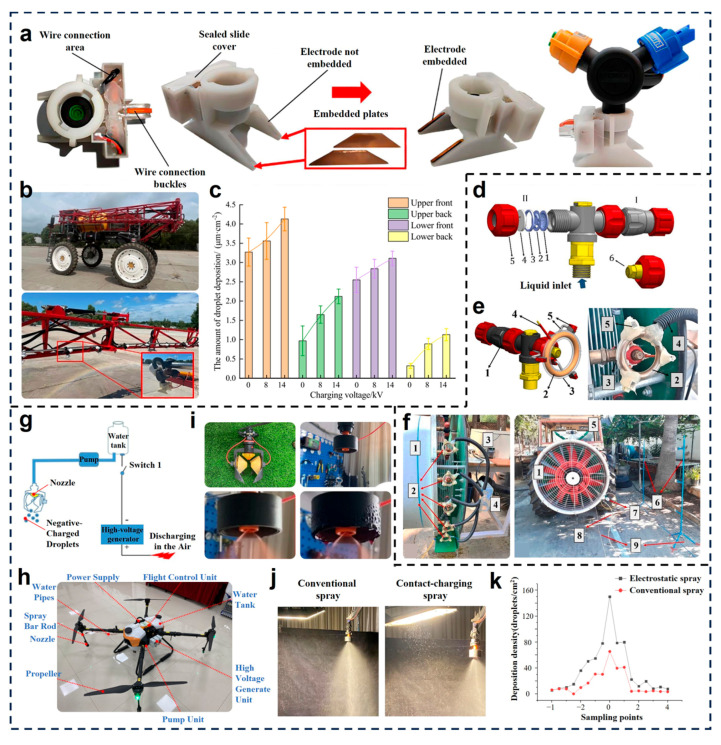
Integration of intelligent spraying systems with electrostatic modules. (**a**) Inductive electrostatic spray device with electrode plate embedded [75]. (**b**) Induction electrostatic spray system installation site [75]; (**c**) Amount of droplet deposition in different collection areas [75]. Reproduced with permission from Reference [75]. Copyright © 2025, Frontiers Media S.A. (**d**) Hydraulic nozzle used to develop the electrostatic charging units [76]. (**e**) Electrostatic charging unit [76]. (**f**) Air-assisted sprayer with electrostatic charging units and the deposition trials setup [76]. Reproduced with permission from Reference [76]. Copyright © 2024, Elsevier. Contact-charging UAV electrostatic spray system and principle; (**g**) contact-charging principle and (**h**) contact-charging UAV electrostatic spray system [77]. (**i**) Two types of induction charging nozzles and electrostatic phenomenon in ring-shape electrodes inductive-nozzles supplied with overhigh voltage [77]. (**j**) Difference between conventional spray and contact-charging spray [77]. (**k**) Deposition difference between UAV electrostatic spray and conventional spray [77]. Reproduced with permission from Reference [77]. Copyright © 2024, MDPI.

**Figure 4 micromachines-16-01285-f004:**
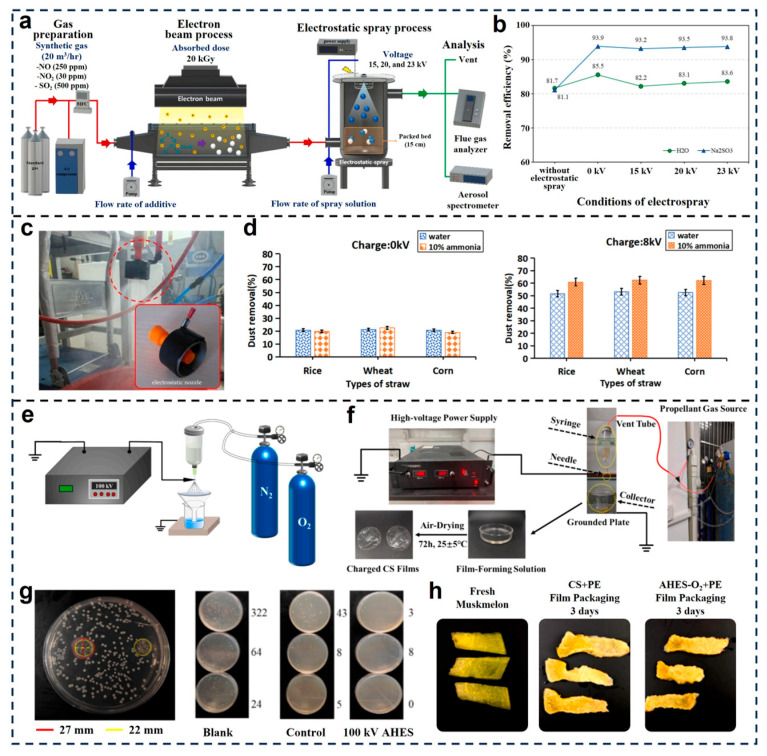
Emerging applications of electrostatic spraying in environmental and material control. (**a**) The configuration of the electron beam-electrostatic spray process [79]; (**b**) NO_X_ Removal Efficiency Under Different Electrostatic Spray Conditions [79]; Reproduced with permission from Reference [79]. Copyright © 2025, Elsevier. (**c**) Electrostatic nozzle [81]. (**d**) Comparison of flue gas dust removal efficiency with and without charge excitation [81]. Reproduced with permission from Reference [81]. Copyright © 2025, MDPI. (**e**,**f**) Schematic of a AHES equipment and the CS film manufacturing process [82]. (**g**) The inhibition zones and the viable bacterial count results of *E. coli*  [82]. (**h**) Comparison of the effects of honeydew melon slices under different packaging methods after 3 days of room temperature storage [82]. Reproduced with permission from Reference [82]. Copyright © 2025, Elsevier.

**Figure 5 micromachines-16-01285-f005:**
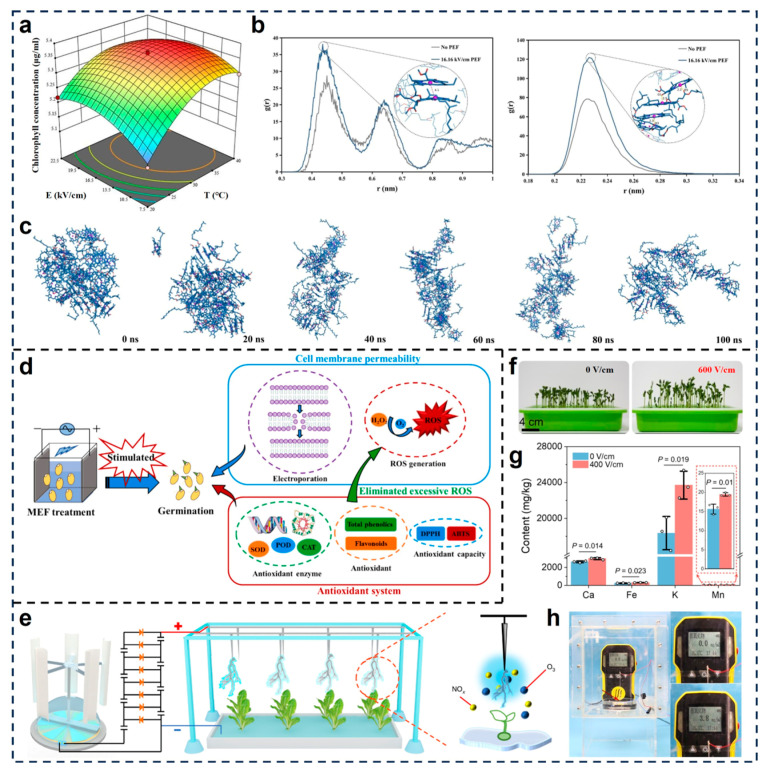
Mechanistic insights into electrostatic stimulation of plant physiology. (**a**) The effects of temperature and electric field intensity on chlorophyll concentration [85]. (**b**) Radial distribution function and the intermolecular interaction of chlorophyll molecules induced by PEF at 16.16 kV/cm (Mg^2+^-Mg^2+^ & chlorophyll O atom-Mg^2+^) [85]. (**c**) 100 ns kinetic simulation snapshot of chlorophyll aggregates under PEF treatment of 16.16 kV/cm [85]. Reproduced with permission from Reference [85]. Copyright © 2025, Elsevier. (**d**) The possible mechanisms by which moderate electric field treatment stimulates brown rice germination [90]; Reproduced with permission from Reference [90]. Copyright © 2025, Elsevier. (**e**) Diagram of the self-powered electric field for pea seedlings treatment [84]. (**f**) Photos of the pea seedlings treated with the self-powered electric field of different strengths for 5 days [84]. (**g**) Trace element (Ca, Fe, K, and Mn) content of the pea seedlings treated with the electric field of different field strengths for 7 days [84]. (**h**) Photograph of the device for the concentration detection of NO_x_ produced by self-powered electric fields [84]. Reproduced with permission from Reference [84]. Copyright © 2022, Spring Nature.

**Figure 6 micromachines-16-01285-f006:**
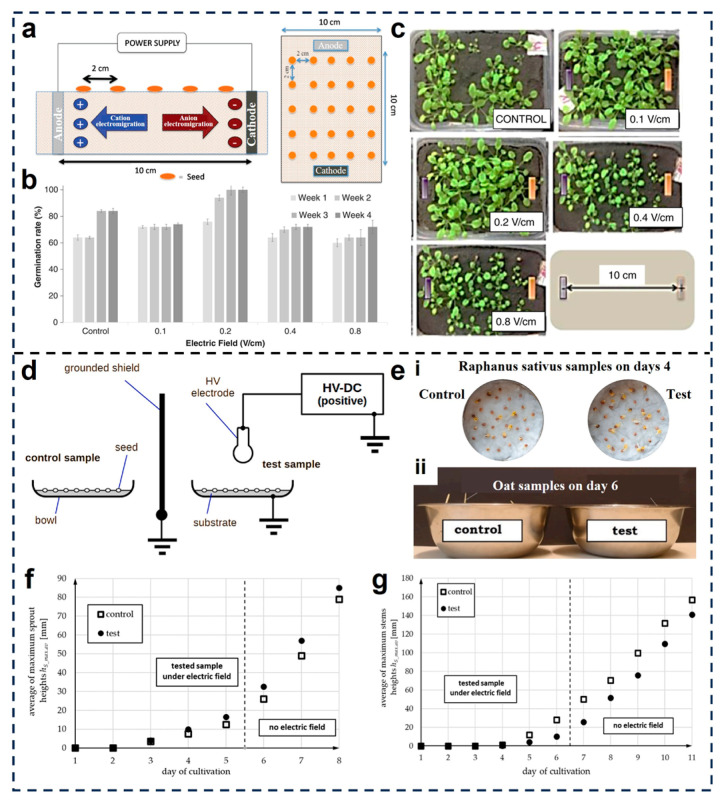
Electrostatic enhancement of seed germination and vigor. (**a**) Schematic representation of electro-culture experiments [92]. (**b**) Germination rate of *A. thaliana* seeds using anodes [92]. (**c**) Rosette diameter of *A. thaliana* plants measured six weeks after electro-culture treatment [92]. Reproduced with permission from Reference [92]. Copyright 2018, WILEY-VCH Verlag GmbH& Co. KGaA, Weinheim. (**d**) Schematic of the test setup [93]. (**e**) Comparison of (**i**) *Raphanus sativus* samples on day 4 and (**ii**) oat samples on day 6 with and without electric field stimulation. The average of maximum stem heights as a function of the day of cultivation for (**f**) *Raphanus sativus* and (**g**) *Avena sativa* [93]. Reproduced with permission from Reference [93]. Copyright © 2024, Elsevier.

**Figure 7 micromachines-16-01285-f007:**
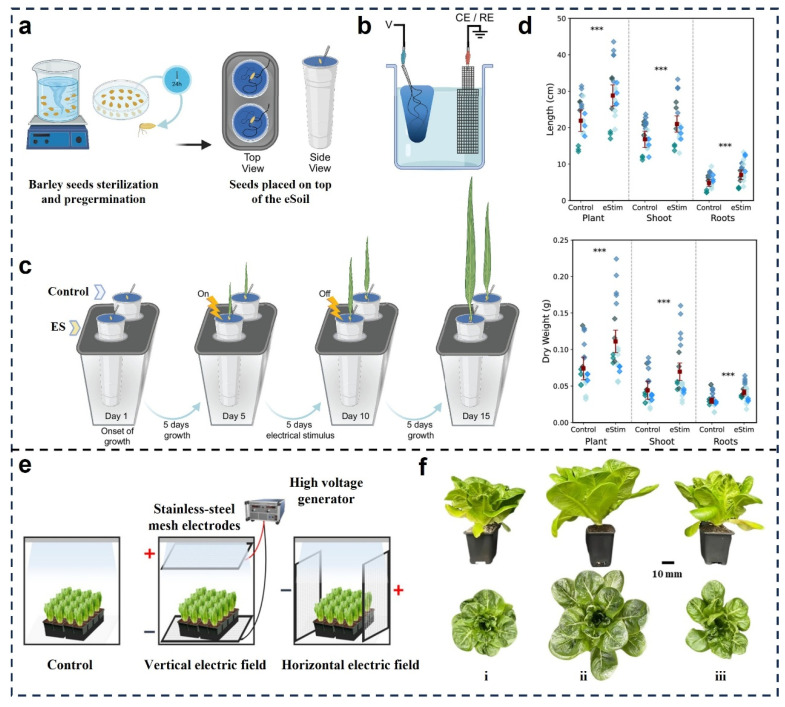
Electrostatic promotion of plant growth and physiological performance. (**a**) Seed sterilization and pregermination protocol [44]. (**b**) Electrical stimulation setup [44]. (**c**) Schematic for the plants’ growth and electrical stimulation protocol [44]. (**d**) Length and dry weight of the plant, shoot, and main root after 15 days of growth in eSoil with and without electrical [44]. *** *p* value < 0.005. Reproduced with permission from Reference [44]. Copyright © 2024, National Academy of Sciences. (**e**) Experimental system for electric fields with different directions [94]. (**f**) Overview and top shot of lettuce plants subjected to electric fields with different directions for 28 days [94]. (i) Control; (ii) Vertical electric field; (iii) Horizontal electric field. Reproduced with permission from Reference [94]. Copyright © 2024, Spring Nature.

**Figure 8 micromachines-16-01285-f008:**
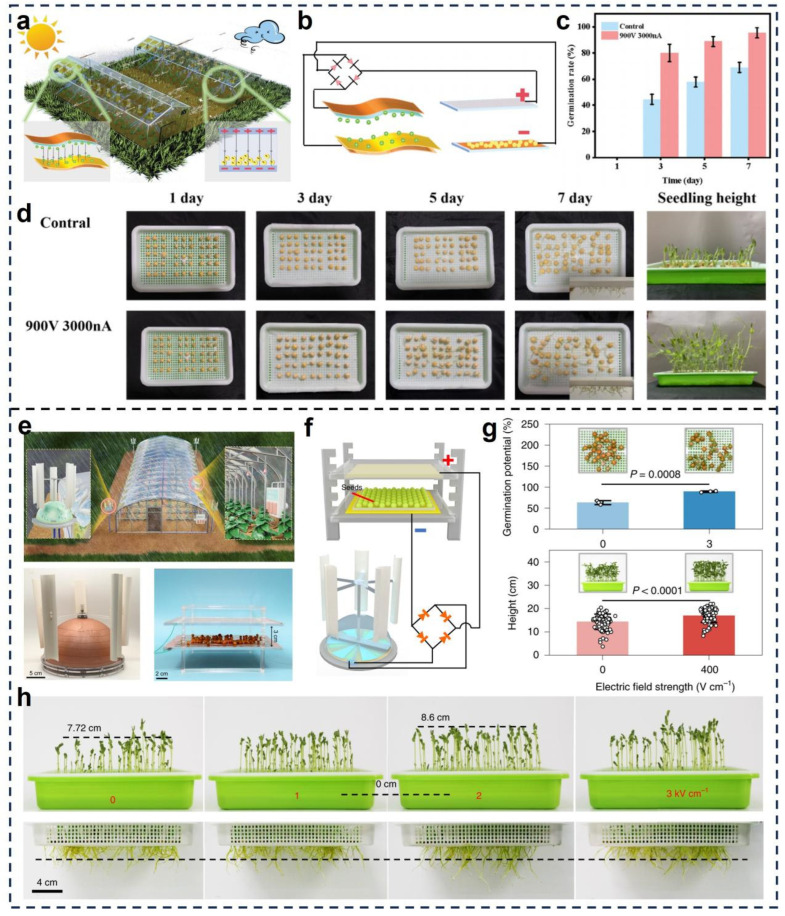
TENG-Based Self-Powered Stimulation Systems. (**a**) Schematic diagram of the practical application scenario [107]. (**b**) Installation diagram of the TENG [107]. (**c**) The germination rates of peas seeds in seven days under 900 V and 3000 nA [107]. (**d**) Germination photos of peas seeds within seven days under TENG and control group [107]. Reproduced with permission from Reference [107]. Copyright 2025, WILEY-VCH Verlag GmbH& Co. KGaA, Weinheim. (**e**) Schematic diagram of the self-powered electrical stimulation system and the Photographs of the TENG and electric field generators for germination promoting [84]. (**f**) Schematic representation of the self-powered electric field for seed treatment [84]. (**g**) The positive effect of the self-powered electrical stimulation system on germination and growth of peas [84]. (**h**) Photographs of the stems and roots of the peas treated with the self-powered electric field of different strengths after 5 days of germination [84]. Reproduced with permission from Reference [84]. Copyright © 2022, Spring Nature.

## Data Availability

No new data were created or analyzed in this study.

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
