# Peer review of "Advances in the Application of Electrostatics in Agriculture: A Review from Macroscale Spray Engineering to Microscale Plant Biostimulation"

_micromachines, 2025, doi:10.3390/mi16111285_

Round 1

Reviewer 1 Report

Comments and Suggestions for Authors

Dear Authors,

The thematic approach of the manuscript is both relevant and innovative, and its overall structural organization is sound. However, the introductory chapter would benefit from improved clarity through stylistic refinement of certain sentences and greater lexical variation to avoid repetition of expressions such as “precise delivery” and “non-contact manipulation.” The transition between the theoretical overview and the classification framework requires a smoother shift, ideally supported by an additional sentence that links the underlying physical principles to the structure of the review. The chapter on macroscale applications is technically comprehensive, yet textual clarity diminishes due to overly long sentences that combine multiple layers of information without rhythmic relief. Repetition of terms such as “droplet deposition efficiency,” “electrostatic induction,” and “uniformity” further burdens the style. Transitions between subsections appear abrupt, lacking introductory sentences that would guide the reader, while the chapter conclusion is missing a synthetic statement that connects macroscale spraying to microscale biostimulation and introduces the subsequent section. The microscale section is informative but overloaded with extended sentences and loosely connected examples. It would benefit from the inclusion of introductory and concluding sentences that thematically link the studies and enhance cohesion. The discussion is conceptually strong, yet the final sentence remains incomplete and should be supplemented with a clear statement emphasizing the importance of interdisciplinary collaboration.

The manuscript has the potential to become a valuable contribution to the field of agricultural electrostatics, provided that stylistic revisions, technical corrections, and stronger thematic integration of concluding passages are implemented.

Comments on the Quality of English Language

The English in the manuscript is functional and technically correct, but it requires stylistic improvements to more clearly convey the research messages and enhance readability. In several sections, overly long sentences combine technical descriptions, results, and interpretations without rhythmic relief, which makes comprehension more difficult. Repetition of expressions (e.g., “precise delivery,” “electrostatic induction,” “germination rate”) contributes to linguistic monotony and could be avoided through the use of synonyms or paraphrasing. Grammatical and punctuation errors are sporadic but present—especially in sentence structure and spacing. The terminology is generally appropriate, but certain expressions could be more precisely defined in the context of the target audience. With language editing and stylistic optimization, the text would significantly improve in clarity, flow, and professional tone.

Reviewer 2 Report

Comments and Suggestions for Authors

The list of comments is attached.
